# Microstructural Evaluation of Thermal-Sprayed CoCrFeMnNi$_{0.8}$V High-Entropy Alloy Coatings

**Athanasios K. Sfikas** [1,*], **Spyros Kamnis** [2], **Martin C. H. Tse** [3], **Katerina A. Christofidou** [3], **Sergio Gonzalez** [1], **Alexandros E. Karantzalis** [4] and **Emmanuel Georgatis** [4]

[1] Faculty of Engineering and Environment, Northumbria University, Newcastle upon Tyne NE1 8ST, UK; sergio.sanchez@northumbria.ac.uk

[2] Castolin Eutectic-Monitor Coatings, Newcastle upon Tyne NE29 8SE, UK; spyros.kamnis@castolin.com

[3] Department of Materials Science and Engineering, The University of Sheffield, Sheffield S1 3JD, UK; chmtse1@sheffield.ac.uk (M.C.H.T.); k.christofidou@sheffield.ac.uk (K.A.C.)

[4] Department of Materials Science and Engineering, University of Ioannina, 45110 Ioannina, Greece; akarantz@uoi.gr (A.E.K.); mgeorgat@uoi.gr (E.G.)

[*] Correspondence: thanasfi@gmail.com

**Abstract:** The aim of this work is to improve the understanding of the effect of the cooling rate on the microstructure of high-entropy alloys, with a focus on high-entropy alloy coatings, by using a combined computational and experimental validation approach. CoCrFeMnNi$_{0.8}$V coatings were deposited on a steel substrate with high velocity oxy-air-fuel spray with the employment of three different deposition temperatures. The microstructures of the coatings were studied and compared with the microstructure of the equivalent bulk high-entropy alloy fabricated by suction casting and powder fabricated by gas atomization. According to the results, the powder and the coatings deposited by low and medium temperatures consisted of a BCC microstructure. On the other hand, the microstructure of the coating deposited by high temperature was more complex, consisting of different phases, including BCC, FCC and oxides. The phase constitution of the bulk high-entropy alloy included an FCC phase and sigma. This variation in the microstructural outcome was assessed in terms of solidification rate, and the results were compared with Thermo-Calc modelling. The microstructure can be tuned by the employment of rapid solidification techniques such as gas atomization, as well as subsequent processing such as high velocity oxy-air-fuel spray with the use of different spray parameters, leading to a variety of microstructural outcomes. This approach is of high interest for the field of high-entropy alloy coatings.

**Keywords:** high-entropy alloy coatings; thermal spray; gas atomization; cooling rate

## 1. Introduction

High-entropy alloys (HEAs) are an emerging category of metallic materials consisting of at least five elements, with the concentration of each element between 5 and 35 at% [1]. The core effects of HEAs include high entropy, highly distorted lattice, sluggish diffusion effect and the cocktail effects [2]. HEAs have attracted a lot of attention due to their properties, such as good mechanical performance [3] and high resistance to surface degradation [4,5].

The equimolar CoCrFeMnNi is one of the most widely studied HEAs. It consists of a single-phase face-centred cubic (FCC) solid solution [6] exhibiting low strength and high ductility [7]. Vanadium addition in CoCrFeMnNi has a profound effect on the microstructure and the mechanical properties. Whilst small additions of V do not lead to a change in the microstructure, as the V content increases, there is a transition in the microstructure from a single FCC solid solution to a dual phase microstructure of FCC and sigma intermetallic phase [8]. This has been explained in terms of poor compatibility between V and the other alloying elements that causes significant distortion to the solid solution crystal lattice, leading to transformation to sigma [7]. V is a stronger sigma-forming element

than Cr; however, the overall concentration of both elements is crucial for determining the volume fraction of sigma in CoCrFeMnNiV systems. Introduction of sigma leads to increased strength and decreased ductility in CoCrFeMnNiV-based systems. As a result, the optimization of the volume fraction of sigma is crucial in order to achieve high strength without compromising ductility [7,8]. Unsolicited contamination in CoCrFeMnNiV fabricated by mechanical alloying and spark plasma sintering may lead to the suppression of sigma and favour the formation of Cr rich carbides and Cr-V rich oxides in an FCC solid solution matrix. This has been explained in terms of the preferential carbide and oxide formation that depletes the alloy from Cr and V, decreasing their bulk concentration such that their content is not sufficient to form a sigma phase [9].

Thermally sprayed HEA coatings have attracted a lot of attention in recent years due to their good surface degradation properties [10,11]. HEA coatings have been applied with the use of different thermal spray techniques, including detonation spraying [12], atmospheric plasma spraying [13–15], high velocity oxy–fuel spraying [16,17] and high velocity air–fuel spraying [17]. Different types of powder have been employed, including gas atomized [12,16] and mechanically alloyed [13,15]. In several cases the coatings retained the phase constitution of the powder [17] while in other cases additional phases were introduced during the application of the coating [13,15]. According to Lobel et al., thermal input is an important parameter affecting the microstructure during spraying [17].

Solidification rate is an important parameter that may affect the microstructure and the properties of HEAs. Chen et al. studied the effect of the cooling rate on the microstructure and mechanical properties of a vacuum-arc-melted $CrFeCoNiAl_{0.6}$ that was subsequently remelted in copper moulds of various sizes (i.e., 2, 3, 5, and 7 mm diameter). According to the findings, control of the cooling rate leads to microstructural optimization through the control of the fraction of the body-centred cubic (BCC) phase and the refinement of FCC, allowing for the tuning of the mechanical properties [18]. He et al. investigated the kinetic effect on the phase transformation and selection of a $CoCrFeNiTi_{0.4}$ HEA. Under low cooling rate, the microstructure consists of multiple phases including FCC, sigma, R and $\gamma'$ However, rapid solidification led to the suppression of the formation of the intermetallic phases of sigma, R and $\gamma'$. It was deduced that solidification rate and solid-state transformation kinetics could be used to control the phase-selection of HEAs for improved performance [19]. In another recent work on an $Al_{0.3}CoCrFeNiMo_{0.75}$ HEA it was concluded that powder produced through inert gas atomization exhibited different phase constitution compared to the equivalent vacuum-arc-melted material. The metastable microstructure of the powder was retained after HVOF spray, influencing the wear behaviour of the coating [20].

To the best of the authors' knowledge, the literature on $CoCrFeMnNi_{0.8}V$ coatings is rather limited. In order to improve the understanding of the microstructural evolution in the $CoCrFeMnNi_{0.8}V$ system, a binary approach was selected, including Thermo-Calc modelling studies and experimental validation. The experimental validation focuses on studying the microstructure of $CoCrFeMnNi_{0.8}V$ HEA in different configurations: gas atomized powder, high velocity oxy-air-fuel (HVOAF) coatings applied by different temperatures, and suction-cast bulk material. A primary goal is to assess how the different cooling rates employed in different processing routes affect the microstructure. Furthermore, this study aims to understand how the coating microstructure is affected by subsequent heat treatment. Another goal is to study the microstructure of $CoCrFeMnNi_{0.8}V$ coatings applied by HVOAF with the use of different spraying temperatures. The CoCrFeMnNiV system is based on the widely studied cantor alloy and has high potential for uses in multiple demanding applications due to its high hardness and high wear-resistance. According to the literature, bulk CoCrFeMnNiV is 4.5 harder than CoCrFeMnNi [8]. Additionally, bulk $CoCrFeMnNi_{0.8}V$ exhibits six-times higher scratch hardness compared to the cantor alloy, indicating the high wear resistance of the material [21]. It can be thus concluded that the system may find use in applications demanding high wear-resistance. In this work,

the Ni content was lowered to increase hardness, since Ni stabilises the soft FCC solid solution [22].

## 2. Materials and Methods

### 2.1. Fabrication of Bulk Samples

CoCrFeMnNi$_{0.8}$V bulk samples were fabricated with the use of pieces of raw elements of high purity (99.9 wt%) with the employment of an Edmund Buhler MAM-1 compact arc-melting apparatus. Proper quantities of the raw materials were measured with the use of a Fisherbrand analytical balance and then ultrasonically cleaned. Afterwards, they were inserted in the furnace chamber and melted under Ar to acquire the master alloy. A Ti getter was used to reduce oxidation. The master alloy was suction-cast into a water-cooled Cu mold with diameter of 8 mm and length of 30 mm. The cylindrical samples were left to cool in the furnace. Afterwards, they were cut into smaller pieces and were subject to standard metallographic preparation procedures to study the microstructure.

### 2.2. Coatings Deposition

CoCrFeMnNi$_{0.8}$V powder was fabricated with gas atomization; it is an R&D product of Castolin Eutectic. The spray setup was the same as that described in our previous works [16,23], with an HVOAF gun specially designed to spray a variety of materials. The torch used a mixture of hydrogen (up to 400 SLPM), air (up to 600 SLPM) and oxygen (up to 300 SLPM), depending on the mode of operation. The combustion chamber pressure varied from 6 to 9 bar. To achieve different flame temperatures, and in turn different coating deposition temperatures, three sets of spray flow rate parameters were utilized and labelled as high temperature (HT), medium temperature (MT) and low temperature (LT). These flow-rate regimes affected the combustion process within the HVOAF gun, with the air flow rate being particularly important as it played a role in decreasing the flame temperature by only partially participating in the combustion process. At HT, the maximum flame temperature was 2500 K, while for MT and LT conditions the simulated combustion temperatures were 2000 K and 1600 K, respectively [16].

The powder was fed using a volumetric disc-based powder feeder, with slight adjustments in RPM and carrier gas flow-rate to keep the powder feed rate constant in all runs. The coatings were deposited on S275 steel plates (nominal composition: C < 0.25 wt%, Mn < 1.6 wt%, S < 0.05 wt%, P < 0.04 wt%, Si < 0.05 wt%) of 50 × 50 mm and 6 mm thickness traversing the gun linearly (raster scan pattern) using a robotic arm at 500 mm/s with a step size of 2 mm. The spray angle was fixed at 90 degrees and the gas flow rates were controlled using a digital console. The LT coating was cut into smaller pieces of 1 cm$^2$ and was subjected to heat treatment in an argon-containing furnace. The samples were heat treated for different periods of time (8, 24, 72 h) at 500 °C to assess the microstructural stability of the system.

### 2.3. Materials Characterization

The suction cast samples, the gas atomized powder and the coatings were studied with the employment of a Rigaku Smartlab SE X-ray diffractometer (Cu-Ka radiation, standard split) and a Tescan Mira 3 scanning electron microscope equipped with an Oxford Instruments EDS analyser. EDX data presented in the manuscript for the different configurations and phases are the average of three measurements. X-ray diffractograms were acquired with a step size of 0.01 °/s. Hardness measurements were conducted for the suction cast samples with the employment of a Wilson VH1150 macro-Vickers hardness tester (Indentation load 196.1 N, holding time 10 s).

### 2.4. Thermo-Calc Modelling

Thermodynamic calculations were performed using Thermo-Calc software, coupled with the SSOL4 solid solution database. The SSOL4 database was used to calculate the one-dimensional phase diagram and to carry out solidification simulations. Solidification

simulations were carried out using both the classic Scheil as well as the Scheil-with-solute trapping models, with solidification speeds between 1 m/s and 10 m/s.

## 3. Results and Discussion

Figure 1 presents the X-ray diffractograms of the suction-cast bulk $CoCrFeMnNi_{0.8}V$, the gas atomized powder and the HEA coatings deposited at different temperatures (LT, MT, HT) by HVOAF. According to the results, the bulk material (Figure 1a) corresponds to the sigma intermetallic phase and an FCC phase. In contrast, the gas atomized powder (Figure 1b), LT (Figure 1c) and MT (Figure 1c) coatings consist of a single BCC phase. In the case of the HT coating (Figure 1c), the main peaks correspond to BCC, but peaks of lower intensity correspond to FCC and oxides. Heat treatment of the LT coating at 500 °C for up to 72 h (Figure 1d) did not lead to any significant changes in the X-ray diffractograms. The coating retained the phase constitution.

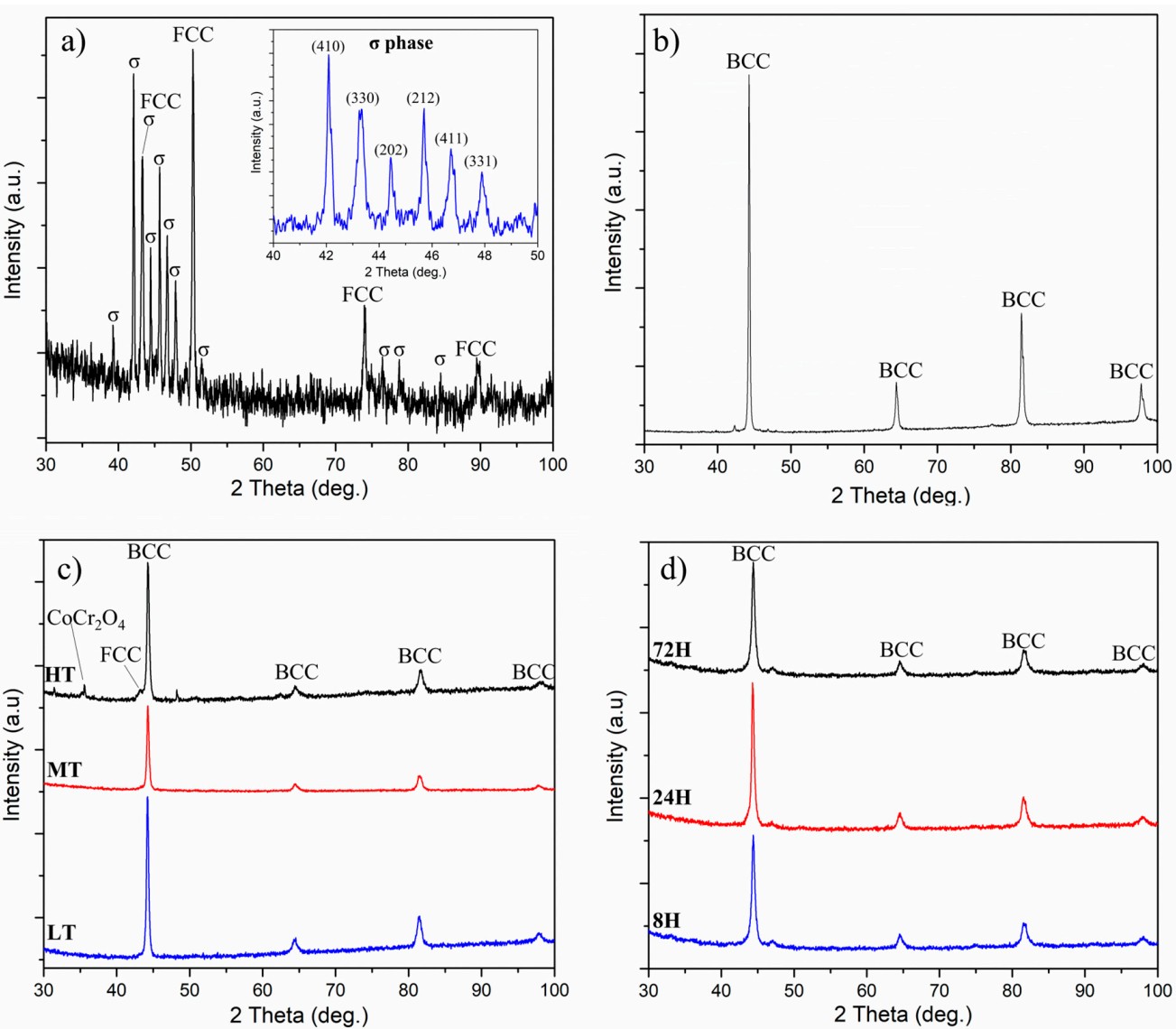

**Figure 1.** X-ray diffractograms for $CoCrFeMnNi_{0.8}V$: (**a**) suction cast bulk material (inset: magnification of the XRD peaks for σ phase), (**b**) gas atomized powder, (**c**) HVOAF coatings applied with different spray temperatures (LT, MT, HT), (**d**) HVOAF coating (LT) after heat treatment at 500 °C (8 h, 24 h, 72 h).

Sigma phase has a tetragonal crystal structure (Space group 136:P 4 2/mnm) [24] with lattice parameters a = b = 0.8844 nm and c = 0.45935 nm, close to those reported by other authors for steel [25]. To better understand the effect of the cooling rate, FCC and BCC phases have been characterized from the XRD scans (Figure 1) in terms of lattice parameters and crystallite sizes. The XRD peaks for the FCC phase in the suction-cast bulk material (Figure 1a) have been detected at the following angles: 42.09° (111), 50.27° (200), 73.97° (220) and 89.57° (311); the calculated lattice parameter is 3.648 Å. From the Scherrer relation, the crystallite size obtained is 24.656 nm. The BCC gas atomized powder (Figure 1b) consists of XRD peaks at 44.27° (110), 64.39° (200), 81.50° (211) and 97.81° (220), for which the calculated lattice parameter is 2.891 Å and the crystallite size is 27.120 nm. For the different spray temperatures, the lattice parameter and the crystallite size of the BCC phase is the following: For LT coating (Figure 1c), the XRD peaks are detected at 44.28° (110), 64.57° (200), 81.61° (211) and 97.93 (220), for which the calculated lattice parameter is 2.16456 Å and the crystallite size is 26.488 nm. For the MT coating (Figure 1c), the XRD peaks were detected at 44.23° (110), 64.55° (200), 81.56° (211) and 98.18° (220), for which the calculated lattice parameter is 2.887 Å and the crystallite size is 19.128. For the HT coating (Figure 1c), the XRD peaks are detected at 44.31° (110), 64.51° (200), 81.81° (211) and 98.06° (220), for which the calculated lattice parameter is 2.886 Å and the crystallite size is 22.630 nm.

XRD data analysis needs to take into consideration that a higher cooling rate results in a decrease of grain size and increase of lattice parameter. The increase of the lattice parameter happens because lattice is further from the relaxed equilibrium conditions. The lattice parameter of the BCC phase of MT and HT is practically the same and larger than the one for LT coating, thus suggesting that the cooling rate for MT and HT coatings is higher than LT. This confirms that the crystallite lattice for the coatings deposited with the employment of LT is in a more relaxed state as compared to that deposited with HT. The faster cooling rate for MT and HT conditions are consistent with the grain size of the BCC phase, close to 20 nm, but for the LT coating the grain size is larger, about 26 nm. For the suction-cast bulk material, the FCC crystallite is 24.656 nm, smaller than the 27.120 nm for the BCC phase.

The examination of the CoCrFeMnNi$_{0.8}$V fabricated by suction casting under SEM indicates that the HEA consists of two phases (Figure 2), a sigma phase (light grey phase) and an FCC solid solution (dark grey phase). Sigma is enriched in Cr and V and depleted in Mn and Ni (Figure 2, Table 1). In contrast, the FCC solid solution is enriched in Mn and Ni and depleted in Cr and V (Figure 2, Table 1). The formation of sigma is expected, since Cr and V are sigma-forming elements, with V being the most potent in the formation of sigma [8]. Stepanov et al. reported a dual sigma-and-FCC microstructure in a CoCrFeMn-NiV HEA fabricated by vacuum arc melting, in good agreement with this study [8]. Closer observation of the FCC phase indicates the presence of another phase developed within. This phase is rich in V and is very fine. The hardness of suction-cast CoCrFeMnNi$_{0.8}$V is 753 HV, approximately six times higher than the hardness of CoCrFeMnNi (128 HV), and can be explained by the high hardness of the prevailing sigma intermetallic phase and the increased hardness of the FCC solid solution due to the increased solid solution strengthening from the introduction of V. The V-rich phase will also have a beneficial role in strengthening the material as well, albeit the exact strengthening derived from these mechanisms was not specifically investigated in this work.

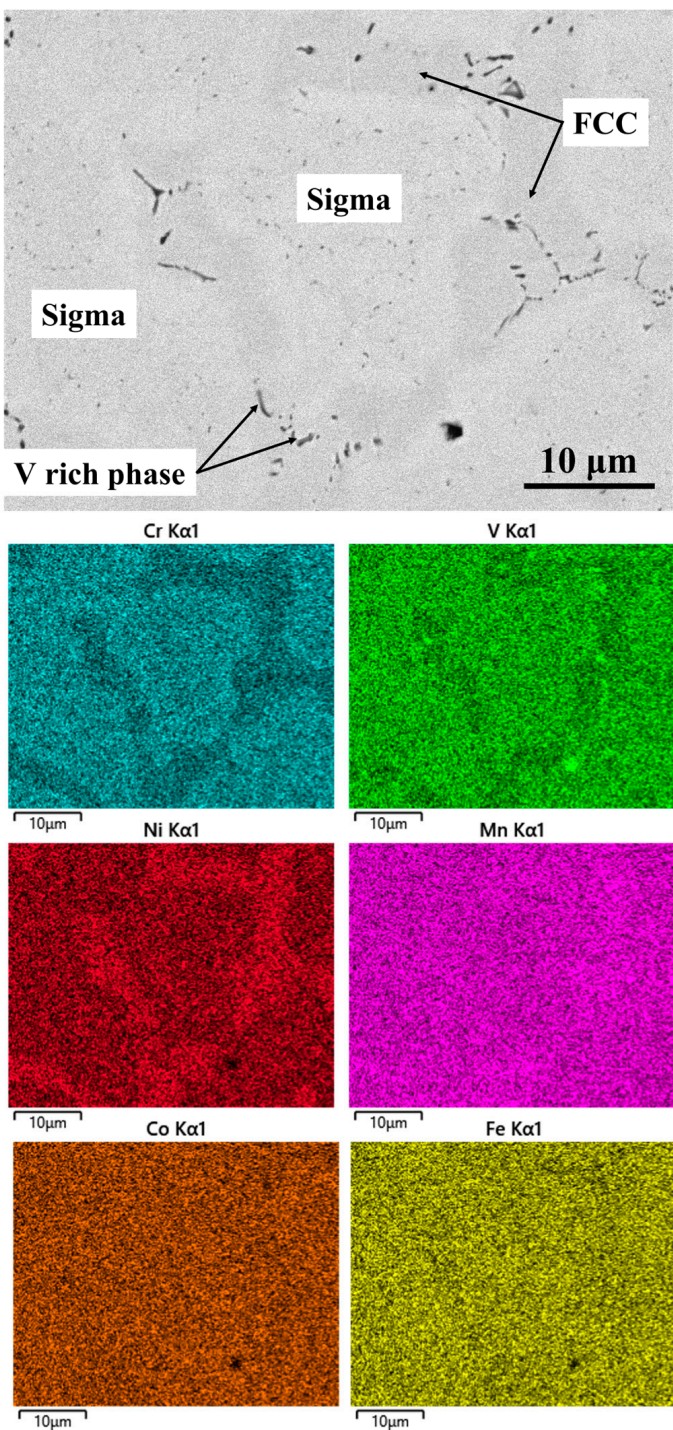

**Figure 2.** Microstructure (BSE mode) and elemental map of CoCrFeMnNi$_{0.8}$V fabricated by suction casting.

**Table 1.** Chemical composition (at%) of bulk CoCrFeMnNi$_{0.8}$V fabricated by suction casting.

|          | V            | Cr           | Mn           | Fe           | Co           | Ni           |
|----------|--------------|--------------|--------------|--------------|--------------|--------------|
| Nominal  | 17.2         | 17.2         | 17.2         | 17.2         | 17.2         | 13.8         |
| Measured | 16.6 ± 0.8   | 19.1 ± 3.5   | 18.2 ± 1.8   | 15.7 ± 1.3   | 17.4 ± 2.1   | 13.1 ± 1.2   |
| Sigma    | 15.9 ± 0.7   | 19.8 ± 2.6   | 16.8 ± 1.9   | 16.1 ± 0.6   | 18.8 ± 1.2   | 12.6 ± 0.7   |
| FCC      | 14 ± 1.6     | 14.3 ± 2.4   | 21.7 ±2.3    | 14.7 ± 1.4   | 18.2 ± 0.2   | 17.1 ± 3.1   |

Gas-atomized CoCrFeMnNi$_{0.8}$V particles consist of a single BCC solid solution with an average size of 20–50 μm (Figures 1b and 3a). The particles appear to have been properly melted and solidified with no porosity and minimal defects (Figure 3b). The chemical composition of the powder is close to the nominal composition of the system, with slightly lower V content and higher Cr content (Table 2). Elemental mapping of the powder indicates the good distribution of the elements, with Ni appearing to be segregated (Figure 4).

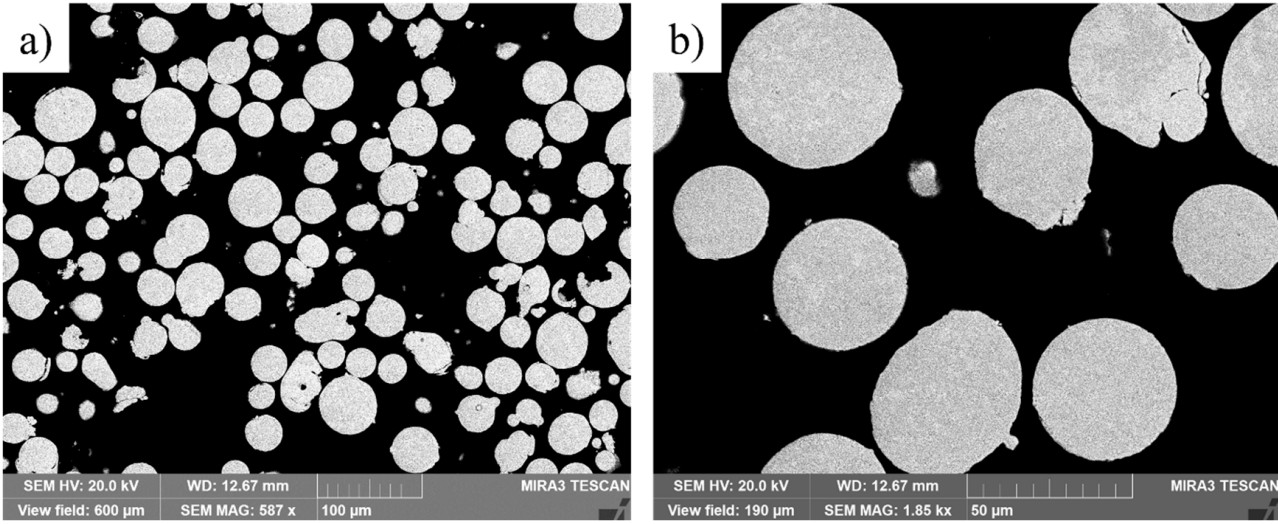

**Figure 3.** Microstructure (BSE mode) of CoCrFeMnNi$_{0.8}$V powder: (**a**) low magnification, (**b**) high magnification.

CoCrFeMnNi$_{0.8}$V coatings were successfully sprayed on a carbon steel substrate by HVOAF with the employment of three different temperatures (Figure 5). The coatings exhibited good distribution of the elements, without any signs of segregation (Figure 5). Regardless of the application temperature, the elemental map indicates that the coatings did not exhibit substantial oxidation during the deposition. Furthermore, according to the X-ray diffractograms of the coatings (Figure 1c), only the HT coating had a minor peak corresponding to oxides, another indication of the low oxides content of the coatings. This is an important finding, indicating that the system shows low sensitivity to the spray temperature when the HVOAF technique is used. However, in a previous work by the authors, an HVOAF-sprayed CoCrFeMnNi coating exhibited high sensitivity to the application temperature [16]. In more detail, as to the CoCrFeMnNi coating, it was indicated that the higher the application temperature, the higher the oxide content. A likely explanation for this discrepancy is that CoCrFeMnNi$_{0.8}$V coatings benefit from the presence of V, which improves the high temperature oxidation of the system. Vanadium is known to exhibit chromium-like behaviour in alloys, meaning that it can form stable oxide films that are resistant to further oxidation. This improves the high-temperature oxidation resistance of the alloy. In relation to this study, Vanadium promotes the formation of a protective oxide layer on the surface of the in-flight particles that can act as a barrier to the penetration of oxygen. This oxide layer can also form in the presence of a small amount of oxygen, which means that the oxide layer can form at relatively lower temperatures during the spray (typical for HVOAF applications). Reducing the oxide film depth on the surface of the powder during spray results in lower oxide content in the coating microstructure.

**Table 2.** Chemical composition (at%) of CoCrFeMnNi$_{0.8}$V gas-atomised powder and HVOAF applied coatings (LT, MT, HT).

|  | V | Cr | Mn | Fe | Co | Ni |
|---|---|---|---|---|---|---|
| Nominal | 17.2 | 17.2 | 17.2 | 17.2 | 17.2 | 13.8 |
| Powder | 15.9 ± 0.2 | 18.1 ± 0.2 | 17.9 ± 0.4 | 17.1 ± 0.4 | 17.3 ± 0.2 | 13.8 ± 0.3 |
| LT | 15.8 ± 0.2 | 18.3 ± 0.1 | 17.8 ± 0.2 | 17.2 ± 0.3 | 17.3 ± 0.3 | 13.6 ± 0.4 |
| MT | 15.8 ± 0.2 | 18.4 ± 0.5 | 17.8 ± 0.4 | 16.9 ± 0.3 | 17.2 ± 0.3 | 13.9 ± 0.4 |
| HT | 15.8 ± 0.3 | 18.3 ± 0.3 | 17.6 ± 0.4 | 17.2 ± 0.1 | 17.4 ± 0.4 | 13.7 ± 0.2 |

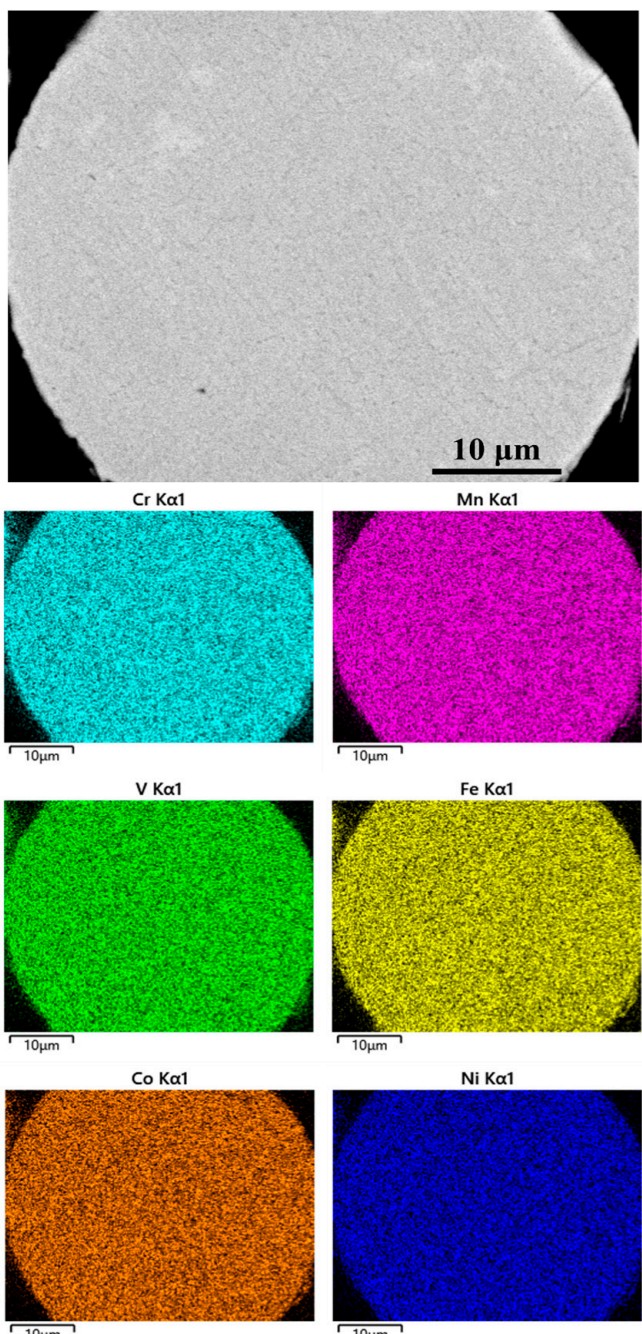

**Figure 4.** Microstructure (BSE mode) and elemental map of CoCrFeMnNi$_{0.8}$V powder.

Higher magnification images of the coatings (Figure 6) and elemental mapping of the HT coating (Figure 7) highlight the differences in the oxidation state for the different

coatings. In more detail, for the HT coating (Figure 6e,f), the periphery of the particles appears to be more oxidized as compared to the LT (Figure 6a,b) and MT (Figure 6c,d) coatings. It should also be noted that the coating particles for the different compositions appear to be deformed in a similar manner. This is attributed to the uniform momentum transfer from the HVOAF process to the inflight particles, under all spray conditions. The chemical composition of the coatings (Table 2) is similar to the chemical composition of the powder, indicating that, due to the low thermal input of HVOAF, elements such as Mn, which have high vapor pressure which may lead to material loss at high processing temperatures, were not depleted.

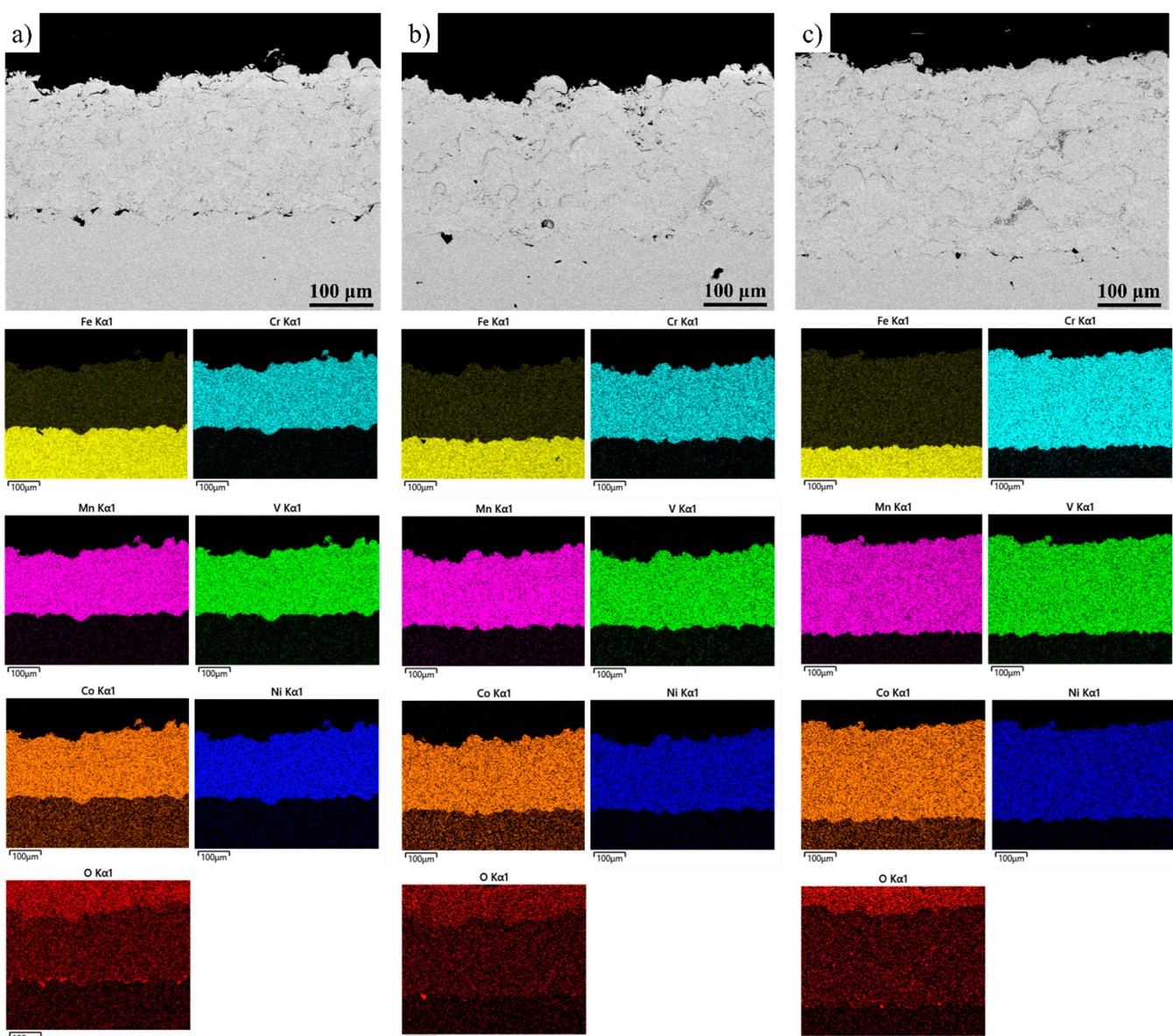

**Figure 5.** Microstructure and elemental map of CoCrFeMnNi$_{0.8}$V coatings deposited under different spraying temperatures: (**a**) LT, (**b**) MT, and (**c**) HT.

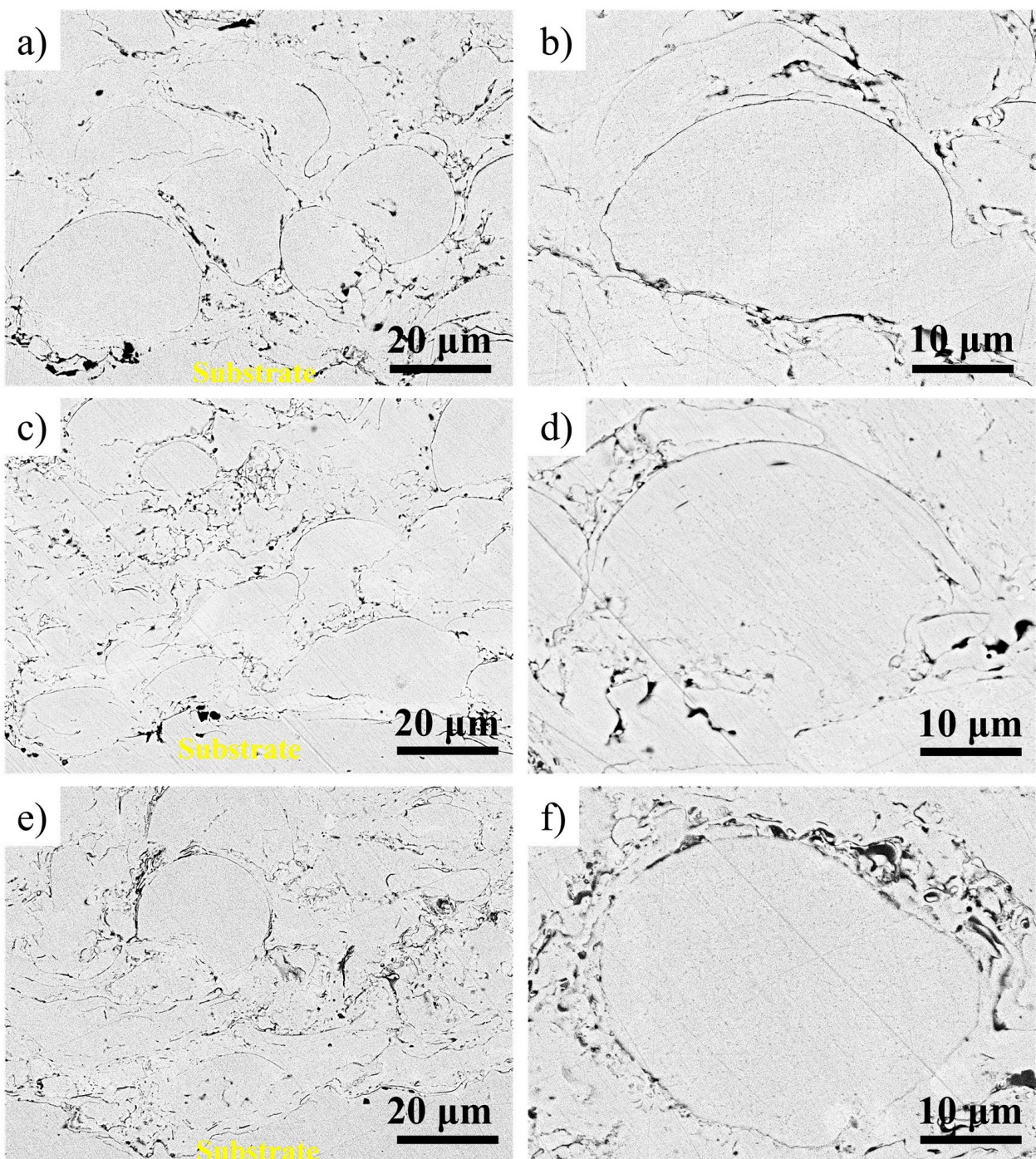

**Figure 6.** Microstructure of CoCrFeMnNi$_{0.8}$V coatings deposited under different spraying temperatures at higher magnification: (**a**,**b**) LT, (**c**,**d**) MT, and (**e**,**f**) HT.

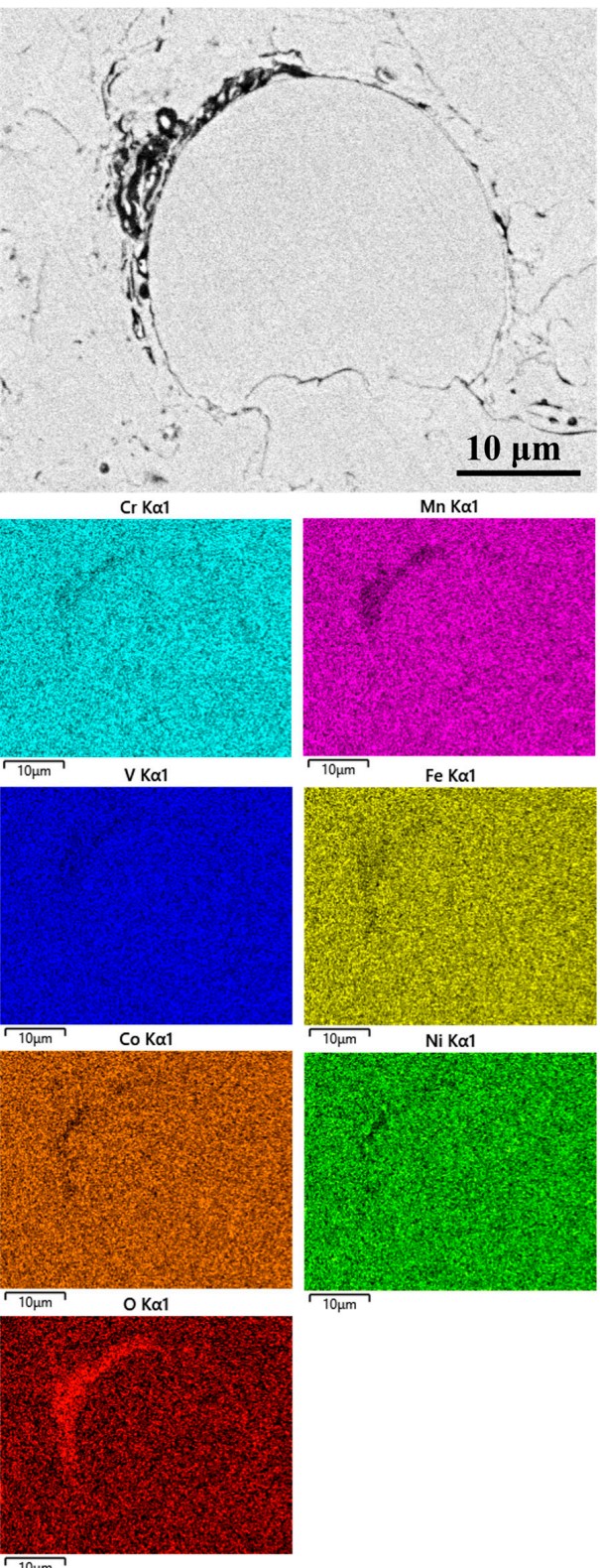

**Figure 7.** Elemental map of the HT CoCrFeMnNi$_{0.8}$V coating.

The one-dimensional phase diagram (Figure 8) provides valuable insight about the equilibrium phase constitution of CoCrFeMnNi$_{0.8}$V. Between the liquidus and 1100 °C, BCC is the only anticipated phase. However, between 400–1100 °C, a dual BCC and FCC microstructure is expected. The volume fraction of FCC increases with decreasing temperatures. Finally, sigma formation is expected at approximately 400 °C.

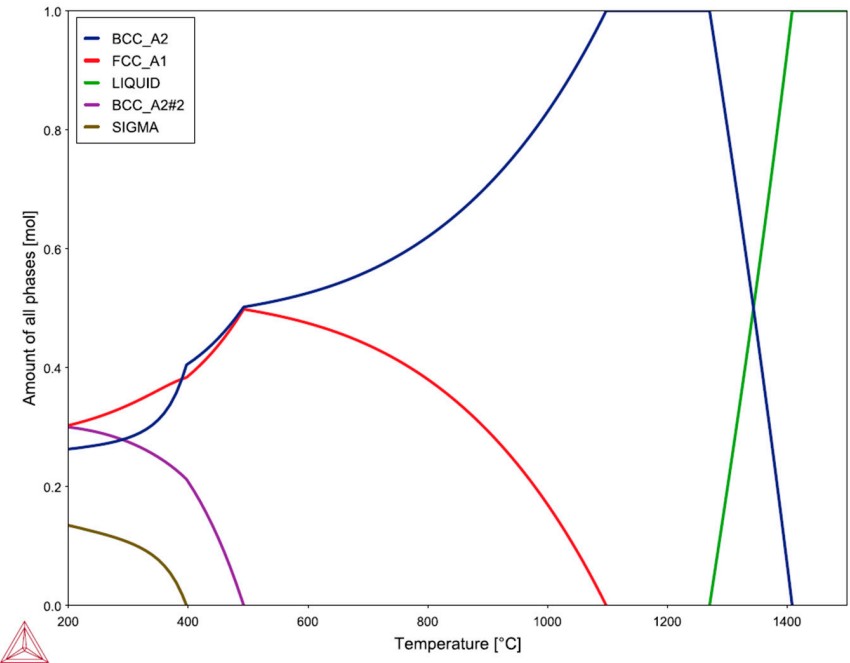

**Figure 8.** One-dimensional phase diagram for CoCrFeMnNi$_{0.8}$V.

Scheil solidification models provide insights into the phases formed at different solidification conditions. The solute trapping model takes into account non-equilibrium conditions from the high cooling rates that are often encountered in gas atomization fabrication methods.

In the Scheil-with-solute trapping models with low to moderate cooling rates (Figure 9, Table 3), it can be observed that the solidification begins with the formation of the BCC phase at all solidification speeds. FCC phase formation follows, resulting in a BCC and FCC microstructure. Compositional analysis calculations of the formed phases (Table 3) showed a trend of increasing V and Cr composition in the resultant BCC phase.

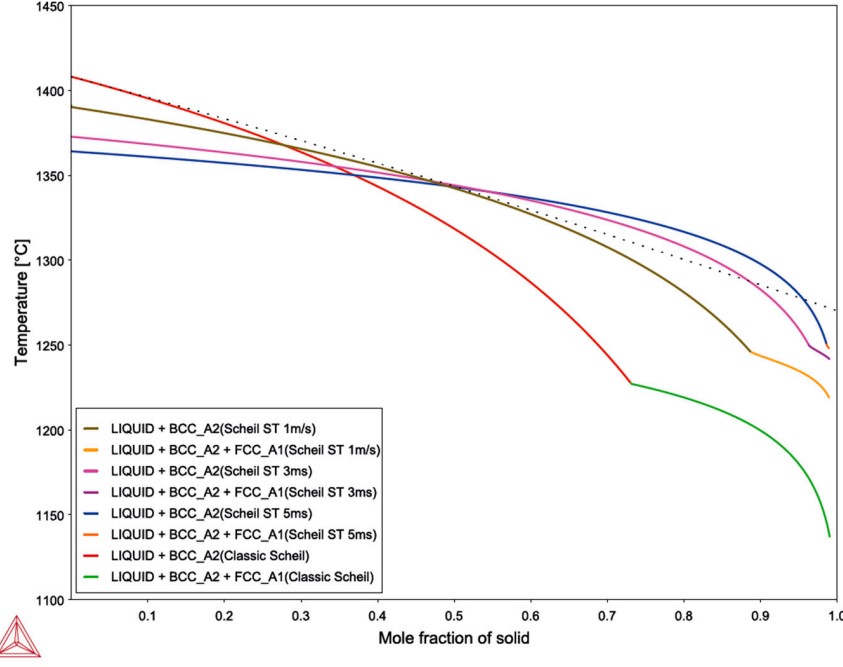

**Figure 9.** Comparison of the solidification phase formations generated using the classic Scheil model, and Scheil-with-solute trapping model with solidification speeds of 1, 3, and 5 m/s.

**Table 3.** Phase constitution for CoCrFeMnNi$_{0.8}$V for different cooling rates at temperatures below liquidus.

| Solidification Rate | Temperature [°C] | Mass Fraction of Solid | Mole Fraction of BCC_A2 | Mole Fraction of FCC_A1 | Mole Percent of V in BCC_A2 | Mole Percent of Cr in BCC_A2 |
|---|---|---|---|---|---|---|
| 1 m/s | 1219.020 | 0.989 | 0.939 | 0.050 | 9.646 | 14.851 |
| 3 m/s | 1241.730 | 0.990 | 0.982 | 0.008 | 8.907 | 16.929 |
| 5 m/s | 1248.080 | 0.989 | 0.989 | 0.000 | 9.137 | 17.090 |
| 7 m/s | 1266.590 | 0.990 | 0.990 | - | 10.756 | 17.307 |
| 10 m/s | 1275.340 | 0.990 | 0.990 | - | 11.639 | 17.368 |

The inconsistency of the microstructure between the bulk CoCrFeMnNi$_{0.8}$V and the gas-atomized equivalent can be explained by analysing the solidification sequence. According to Stepanov et al., during solidification, just below the melting point, the microstructure of CoCrFeMnNiV consists of a BCC and an FCC phase [8]. This was explained by the high content of the HEA in Cr and V that have been reported to stabilize BCC. Upon cooling, the BCC phase, which is rich in Cr and V, transforms into an FCC and sigma through a eutectoid reaction. Stepanov et al. calculated that the volume of BCC in higher temperature stages is approximately 80% for CoCrFeMnNiV [8]. A similar mechanism was observed in a super-duplex stainless steel aged at 920 °C, where the eutectoid dissolution of the delta ferrite led to the formation of secondary austenite and a sigma phase [26]. It is likely that in CoCrFeMnNi$_{0.8}$V, the volume of BCC just below the liquidus is higher than 80% due to the lower content of Ni (CoCrFeMnNi$_{0.8}$V: 13.8 at% Ni, CoCrFeMnNiV: 16.6 at% Ni), which stabilizes the FCC structure in CoCrFeMnNi [22], and the increased volume fraction of Cr and V (CoCrFeMnNi$_{0.8}$V: 34.4 at% V + Cr, CoCrFeMnNiV: 33.3 at% V + Cr) that stabilizes the BCC phase [8]. The metastability of the microstructure can be assessed with the employment of thermodynamic criteria taken from the work of Guo et al. According to this approach, stable FCC phases are formed in nearly equi-atomic HEAs in which valence electron concentrations (VEC) are ≥8, while for VEC < 6.87, stable BCC phases prevail [27]. VEC values for the nominal composition are 7.40, for the suction cast, HEA 7.38, and for the powder, 7.44. Additionally, VEC values for the LT, MT and HT coatings are 7.44, 7.43 and 7.43, respectively. So, according to this criterion, metastable phases are expected in the studied configurations, in good agreement with the experimental observations. According to Tsai et al., in HEAs containing Cr and V, for 6.88 < VEC < 7.84, the formation of sigma is expected [28]. VEC values for all the reported systems fall within this range, so the formation of sigma is expected.

In this effort, the microstructures of CoCrFeMnNi$_{0.8}$V as fabricated by suction casting, gas atomization and HVOAF coatings in sprayed and heat-treated configurations were assessed. Additionally, Thermo-Calc modelling was used to predict the phase constitution under equilibrium conditions and at various solidification rates. The suction-cast HEA consists of a dual FCC and sigma intermetallic microstructure, while the gas-atomized powder and the LT and MT coatings consist of a BCC solid solution. In the case of HT coating, multiple phases are present, including FCC, BCC and oxides. This discrepancy can be explained in terms of solidification rate. In more detail, in suction-casting the cooling rate is approximately 100–1000 K/s [29,30]. On the other hand, the cooling rate during powder manufacturing can be as high as $10^{10}$ K/s [31]. In the suction-cast HEA, during solidification there is a transition from a BCC microstructure in higher temperatures to an FCC and sigma microstructure. In contrast, in the case of gas-powder, rapid solidification inhibits the eutectoid reaction that leads to the formation of FCC and sigma. As a result, the BCC solid solution is the microstructural outcome under rapid solidification. However, this is an off-equilibrium microstructure. The spraying of the powder with HVOAF, especially with LT and MT configurations, leads to a single-phase BCC solid solution microstructure. The microstructure of the HT coatings consists of various phases, indicating that the employment of higher spraying temperature leads to microstructural evolution towards equilibrium. However, despite the high temperature involved in the HT spraying (higher

than the melting point of the powder) the process is very fast, and the phase transformation is incomplete.

This paper highlights the importance of the cooling rate in metastable HEA systems that may lead to off-equilibrium microstructures. In conclusion, the control of the cooling rate is of high importance for the tailoring of the microstructure of metastable HEAs, and can lead to the desired properties. This is of high interest, particularly in HEA coatings where gas atomization, a rapid solidification processing technique, is widely used to produce the powder. The employment of HVOAF is beneficial, since the low thermal input of the technique helps to control microstructural evolution. Another benefit of the use of HVOAF for the CoCrFeMnNiV system is that it may help to keep oxidation at low levels.

## 4. Conclusions

- The aim of this work was to improve the understanding of the effect of the cooling rate on the microstructure of high-entropy alloys, with a focus on high-entropy alloy coatings, by using a combined computational and experimental validation approach.
- In this research effort, $CoCrFeMnNi_{0.8}V$ coatings were deposited under different spray temperatures (i.e., LT, MT and HT) on a steel substrate with the use of gas-atomized powder. For comparison, bulk $CoCrFeMnNi_{0.8}V$ was fabricated by suction casting.
- A variety of microstructural outcomes were observed for the different configurations, including a BCC phase for the gas-atomized powder, LT and MT coatings. On the other hand, the HT coating consisted of a variety of phases including BCC, FCC and oxides. The bulk material had a dual FCC and sigma intermetallic microstructure. Heat treatment for up to 72 h at 500 °C for the LT coating didn't have any significant effect on the microstructure.
- The variation in the observed phases for the different configurations has been explained in terms of solidification rate. In more detail, rapid solidification in gas atomization can inhibit phase transformations, leading to microstructures which are out of equilibrium. Subsequent application of the powder with the use of a low thermal input technique such as HVOAF with the use of different spray parameters can lead to the desired microstructure.
- This approach is of high interest for high-entropy alloy coatings where the raw powders are fabricated by rapid solidification techniques and there are a variety of deposition techniques with a large range of deposition temperatures and parameters.

**Author Contributions:** Conceptualization, A.K.S. and S.K.; methodology, A.K.S., S.K., M.C.H.T. and K.A.C.; validation, A.K.S., M.C.H.T., K.A.C., A.E.K. and E.G.; formal analysis, A.K.S., M.C.H.T. and K.A.C.; investigation, A.K.S., M.C.H.T. and K.A.C.; resources, S.K., S.G. and K.A.C.; data curation, A.K.S. and M.C.H.T.; writing—original draft preparation, A.K.S., S.K. and K.A.C.; writing—review and editing, A.K.S., S.K., K.A.C., S.G., A.E.K. and E.G.; visualization, A.K.S. and S.G.; supervision, S.K. and S.G.; project administration, S.K. and S.G.; funding acquisition, S.K. All authors have read and agreed to the published version of the manuscript.

**Funding:** The authors would like to acknowledge the support of the UK Research and Innovation (UKRI-IUK) national funding agency. Project grant: 53662 "Design of high entropy superalloys using a hybrid experimental based machine learning approach: Steel sector application".

**Institutional Review Board Statement:** Not applicable.

**Informed Consent Statement:** Not applicable.

**Data Availability Statement:** Data available in the paper.

**Conflicts of Interest:** The authors declare no conflict of interest.

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
