# Peer review of "Microstructural Evaluation of Thermal-Sprayed CoCrFeMnNi0.8V High-Entropy Alloy Coatings"

_coatings, doi:10.3390/coatings13061004_

Round 1

Reviewer 1 Report

Authors have already developed the CoCrFeMnNi and studied the coating properties, combined with a Computational and Experimental validation approach [Ref. 16: J. Therm. Spray Technol. 2022, 31, 1000-1010]

What is the significance of the current work from the previously published similar work, except for the addition of Vanadium?

It would be better to provide the effects of coatings on the wear behavior of CoCrFeMnNiV (with AFM images) with different spray temperatures (LT, MT, and HT).

In addition, adhesion behavior is required.

Provide clear images of Fig. 2 and 7 along with EDX spectra.

May remove the elemental map from Fig. 5.

Moderate English correction is required.

Author Response

Authors have already developed the CoCrFeMnNi and studied the coating properties, combined with a Computational and Experimental validation approach [Ref. 16: J. Therm. Spray Technol. 2022, 31, 1000-1010]

What is the significance of the current work from the previously published similar work, except for the addition of Vanadium?

The authors would like to thank Reviewer for the comment. In the manuscript published in J. Therm. Spray Technol. 2022, the authors used a combined computational and experimental validation approach to study the effect of the spray temperature on the velocity, temperature of the powder and the final properties of the coating. In this work the authors have studied how controlling the solidification rate enables to tune the microstructure and phase constitution of HEA coatings. A new approach is proposed, tailoring the microstructure of HEA coatings by producing out of equilibrium microstructures for the powder and then using the optimal parameters during spray to achieve the desired microstructure. Another significant difference is that in the previous publication, a CDF model was used to assess the temperature and the velocity of the powder during spray while in this piece of work Thermo-Calc is used to predict phase formation. There is also a long discussion about the solidification sequence and thermodynamic models that are used to predict phases in HEAs. In addition, in this work we expanded the analysis of the X-ray diffractograms and calculated the lattice parameters and crystallite size. It is worth noticing that the addition of V to the Cantor alloy has an important effect on the microstructure and therefore the on the properties, resulting in a very different alloy system. In conclusion, this work has a very different focus and we have studied a different type of HEA.

It would be better to provide the effects of coatings on the wear behavior of CoCrFeMnNiV (with AFM images) with different spray temperatures (LT, MT, and HT).

In addition, adhesion behavior is required.

The authors would like to thank the Reviewer for the interesting suggestion. This is something we are planning to do in the future (i.e., to assess the corrosion and wear performance of the studied HEA coatings). However, the aim of this manuscript is different. The aim is to produce powders by gas atomization, a rapid solidification technique, that leads to the formation of out of equilibrium microstructures. Using optimal spray parameters with these powders enables to achieve the desired microstructure.

Provide clear images of Fig. 2 and 7 along with EDX spectra.

The authors would like to thank Reviewer for his/her comment. Fig. 2 was replaced with another of higher magnification. The phases have been assigned and highlighted. Fig. 7 was modified to make it easier to distinguish the oxide developed at the periphery of the particle.

May remove the elemental map from Fig. 5.

The authors would like to thank Reviewer for their comment. The authors would like to keep the elemental map shown in Fig. 5 because it provides evidence of the low oxidation of the coatings at the different spraying temperatures. Moreover, this mapping is useful to highlight that thermal-sprayed CoCrFeMnNi0.8V coatings have a small sensitivity to spray temperature. Elemental mapping also highlights that the coatings show no element segregation.

Reviewer 2 Report

In this paper, the effect of the cooling rate on the microstructure of high-entropy alloy coatings by using a combined computational and experimental validation approach. However, the following problems need to be solved before publication:

1.        Please reconsider whether the “keywords” are suitable. In addition, generally speaking, three to five keywords can be selected, and the number of keywords in the article obviously exceeds the requirements.

2.        Page 3, lines 111-118. Do not introduce the HVOAF spraying method in “Materials and Methods”.

3.        It is recommended to divide “Materials and Methods” into subsections so that the structure and content of the article will be clearer.

4.        Page 7, lines 192-193. “The particles appear to have been 192 properly melted and solidified with no porosity or defects”, but is the “cracks” in the upper right corner of the particles a defect or something else in figure 3b?

5.        Page 9, lines 212-243: paragraphs are too long, suggest subparagraphs and restructuring of figures 5 to 7 and the article.

6.        Figure 6 has scratches, can a higher quality figure be provided? It is also suggested to add scale to figure 6 like figure 5.

7.        Figures 8 to 9 are too large, it is recommended to adjust the image size.

8.        Conclusion: the conclusion of this paper is divided into 6 points. Appropriate consolidation is recommended. Moreover, it is necessary to make a general summary at the beginning of the conclusion.

Minor editing of English language required

Author Response

  1. Please reconsider whether the “keywords” are suitable. In addition, generally speaking, three to five keywords can be selected, and the number of keywords in the article obviously exceeds the requirements.

The keywords have been reduced from 6 to 4.

  1. Page 3, lines 111-118. Do not introduce the HVOAF spraying method in “Materials and Methods”.

Thank you for the suggestion. To prevent any confusion, the text about HVOAF in "Materials and Methods" has been removed.

  1. It is recommended to divide “Materials and Methods” into subsections so that the structure and content of the article will be clearer.

The authors would like to thank Reviewer for their suggestion. Materials and methods was divided in four subchapters: fabrication of bulk samples, coatings deposition, materials characterisation and Thermo-Calc modelling. Hopefully, this will make the section easier to read.

  1. Page 7, lines 192-193. “The particles appear to have been 192 properly melted and solidified with no porosity or defects”, but is the “cracks” in the upper right corner of the particles a defect or something else in figure 3b?

       Thank you for the comment. The feature that can be observed in Fig 3b is smaller particles mixing with bigger particles and causing this “satellite” phenomenon. The satellite particles, depending on their size, can be subject to partial or full melting upon introduction in the flame, and may appear to be spherical or deformed. Nonetheless, to be more precise we replaced "no defects" by "minimal defects".

  1. Page 9, lines 212-243: paragraphs are too long, suggest subparagraphs and restructuring of figures 5 to 7 and the article.

Following the suggestion, the large paragraph was split into two smaller paragraphs to make the manuscript easier to read.

  1. Figure 6 has scratches, can a higher quality figure be provided? It is also suggested to add scale to figure 6 like figure 5

To implement your suggestion, the scale on Fig 6 was changed to improve consistency. The authors tried to adjust the images to improve the quality.

  1. Figures 8 to 9 are too large, it is recommended to adjust the image size.

The size of Fig 8. and Fig. 9 was reduced.

  1. Conclusion: the conclusion of this paper is divided into 6 points. Appropriate consolidation is recommended. Moreover, it is necessary to make a general summary at the beginning of the conclusion.

We would like to thank you for the comment. An initial general summary was added to the comments. Furthermore, conclusions were reduced to make the manuscript easier to read.

Reviewer 3 Report

Correctly done and written work about interesting and attractive thematic. Minor suggestions are given in PDF document.

Author Response

Obtained results suggest formation of oxides only at high temperatures. Please explain in more details.

The authors would like to thank Reviewer for his/her comment. The authors wanted to highlight the formation of the oxide at relatively lower spray temperatures involved during HVOF application as compared to the temperatures involved in other techniques like plasma spray. The phrase was changed to improve clarity.

Reviewer 4 Report

The manuscript is interesting, but some explanations and additions should be made.

-The face-centered cubic (FCC) and body-centered cubic (BCC) phases abbreviations should  be defined

-Sigma intermetallic phase should be specified: lattice parameters and space group

-High Velocity Oxygen-Fuel (HVOF) should be defined.

-FCC and BCC phases should be characterized for all samples in diffraction figure 1 in terms of lattice parameters and crystallite sizes using at least the Scherrer relation.

-It would be advisable to discuss if there are differences between the lattice parameters  and crystallite sizes depending on the preparation method and the treatment conditions.

Author Response

-The face-centered cubic (FCC) and body-centered cubic (BCC) phases abbreviations should  be defined

The authors would like to thank Reviewer for his/her suggestion. The abbreviations were defined in the introduction.

-Sigma intermetallic phase should be specified: lattice parameters and space group

The authors would like to thank Reviewer for his/her suggestion. Fig. 1 was modified and an inset for the peaks corresponding to sigma phase was added. Lattice parameters and space group were added:

Sigma phase has a tetragonal crystal structure (Space hroup 136:P 4 2/mnm) [24] with lattice parameters a=b=0.8844 nm and c=0.45935 nm, close to those reported by other authors for steel [25].

-High Velocity Oxygen-Fuel (HVOF) should be defined.

To follow the suggestion, abbreviations were defined in the introduction.

-FCC and BCC phases should be characterized for all samples in diffraction figure 1 in terms of lattice parameters and crystallite sizes using at least the Scherrer relation.

The authors would like to thank Reviewer for his/her suggestion. The lattice parameters and the crystallite sized were calculated by using the Scherrer relation:

To better understand the effect of the cooling rate, FCC and BCC phases have been characterized from the XRD scans (Fig. 1) in terms of lattice parameters and crystallite sizes. The XRD peaks for the FCC phase in the suction cast bulk material have been detected at the following angles: 42.09° (111), 50.27° (200), 73.97° (220) and 89.57° (311) and the calculated lattice parameter is 3.648 Å. From the Scherrer relation the crystallite size obtained is 24.656 nm. The BCC gas atomized powder consists of XRD peaks at 44.27° (110), 64.39° (200), 81.50° (211) and 97.81° (220) for which the calculated lattice parameter is 2.891 Å and the crystallite size is 27.120 nm. For the different spray temperatures, the lattice parameter and the crystallite size of BCC phase is the following: For LT coating the XRD peaks are detected at 44.28° (110), 64.57° (200) and 81.81° (211) for which the calculated lattice parameter is 2.16456 Å and the crystallite size is 19.128 nm. For the MT coating the XRD peaks were detected at 44.23° (110), 64.51° (200), 81.81° (211) and 98.18° (220) for which the calculated lattice parameter is 2.887 Å and the crystallite size is 22.630 nm. For the HT coating, the XRD peaks are detected at 44.31° (110), 64.51° (200), 81.81° (211) and 98.06° (220) for which the calculated lattice parameter is 2.886 Å and the crystallite size is 22.630 nm.

-It would be advisable to discuss if there are differences between the lattice parameters  and crystallite sizes depending on the preparation method and the treatment conditions.

Thank you for the interesting comment. The discussion on the effect of the preparation method and the cooling rate on the lattice parameters and crystallite size was added:

XRD data analysis need to take into consideration that higher cooling rate results in a decrease of grain size and an increase of lattice parameter. The increase of the lattice parameter happens because lattice is further from the relaxed equilibrium conditions. The lattice parameter of the BCC phase of MT and HT is practically the same and larger than the one for LT coating, thus suggesting that the cooling rate for MT and HT coatings is higher. This confirms that the crystallite lattice for the coatings deposited with the employment of LT is in a more relaxed state as compared to the HT. The faster cooling rate for MT and HT conditions are consistent with the grain size of the BCC phase, close to 20 nm but for the LT coating the grain size is larger, about 26 nm. For the suction cast bulk material the FCC crystallite is 24.656 nm, smaller than 27.120 nm for the BCC phase.

Round 2

Reviewer 1 Report

-

-

Reviewer 2 Report

The paper can be accepted now.

The paper is well written.